# Autologous Marrow Mesenchymal Stem Cell Driving Bone Regeneration in a Rabbit Model of Femoral Head Osteonecrosis

**DOI:** 10.3390/pharmaceutics14102127

**Published:** 2022-10-06

**Authors:** Ilenia Mastrolia, Andrea Giorgini, Alba Murgia, Pietro Loschi, Tiziana Petrachi, Valeria Rasini, Massimo Pinelli, Valentina Pinto, Francesca Lolli, Chiara Chiavelli, Giulia Grisendi, Maria Cristina Baschieri, Giorgio De Santis, Fabio Catani, Massimo Dominici, Elena Veronesi

**Affiliations:** 1Laboratory of Cellular Therapy, Division of Oncology, Department of Medical and Surgical Sciences for Children & Adults, University of Modena and Reggio Emilia, 41124 Modena, Italy; 2Division of Orthopedics, Department of Medical and Surgical Sciences for Children & Adults, University Hospital of Modena and Reggio Emilia, 41124 Modena, Italy; 3Technopole of Mirandola TPM, Mirandola, 41037 Modena, Italy; 4Dardano Clinic, Medolla, 41036 Modena, Italy; 5Division of Plastic Surgery, Department of Medical and Surgical Sciences for Children & Adults, University-Hospital of Modena and Reggio Emilia, 41124 Modena, Italy

**Keywords:** osteonecrosis, femoral head, bone marrow, mesenchymal stromal/stem cells, bone regeneration, cell therapy

## Abstract

Osteonecrosis of the femoral head (ONFH) is a progressive degenerative disease that ultimately requires a total hip replacement. Mesenchymal stromal/stem cells (MSCs), particularly the ones isolated from bone marrow (BM), could be promising tools to restore bone tissue in ONFH. Here, we established a rabbit model to mimic the pathogenic features of human ONFH and to challenge an autologous MSC-based treatment. ON has been originally induced by the synergic combination of surgery and steroid administration. Autologous BM-MSCs were then implanted in the FH, aiming to restore the damaged tissue. Histological analyses confirmed bone formation in the BM-MSC treated rabbit femurs but not in the controls. In addition, the model also allowed investigations on BM-MSCs isolated before (ON-BM-MSCs) and after (ON+BM-MSCs) ON induction to dissect the impact of ON damage on MSC behavior in an affected microenvironment, accounting for those clinical approaches foreseeing MSCs generally isolated from affected patients. BM-MSCs, isolated before and after ON induction, revealed similar growth rates, immunophenotypic profiles, and differentiation abilities regardless of the ON. Our data support the use of ON+BM-MSCs as a promising autologous therapeutic tool to treat ON, paving the way for a more consolidated use into the clinical settings.

## 1. Introduction

Osteonecrosis of the femoral head (ONFH), also known as avascular necrosis, is a progressive degenerative disease characterized by a disruption in blood flow to the femoral head (FH) [1]. ONFH is a multifactorial disease caused by traumatic (e.g., interruption of the ligament of teres) or non-traumatic events (e.g., chronic use of alcohol, smoking, or as a result of steroids, radiotherapy, and chemotherapy) [2]. The current treatments are not always effective, and in cases in which the bone architecture collapses, a total hip replacement is required [3]. ONFH can affect relatively young adults with an age at presentation between 20 and 40 years; prosthesis substitution, which occurs approximately every 15 years, contributes to the bone morbidity potentially compromising the quality of life [4]. Thus, alternative therapeutic treatments, possibly using cell-based approaches as performed for other skeletal-related diseases, shall be implemented [5,6,7,8].

In vitro and in vivo models are required to increase our knowledge about the disease and to evaluate the efficacy of new therapies for ONFH. To date, several animal models of ONFH have been generated using surgical or pharmacological approaches [9,10,11,12]. The first consists of the removal of the junctional hip capsule, which is followed by circumferential cauterization of the periosteum and vessels around the FH. The second and more common method is based on the intramuscular injection of methylprednisolone (MPSL), which is a corticosteroid known to be associated with ON. Although ON animal models have been generated, none is able to precisely recreate the human ON pathophysiology. 

In this study, by combining surgical and pharmacological approaches, we originally generated a rabbit ONFH model with features overlapping human ON. This model has been then used for testing the in vivo efficacy of autologous bone marrow mesenchymal stromal/stem cells (BM-MSCs) also accounting for the impact of ON induction in their compartment before (ON-BM-MSCs) and after ON induction (ON+BM-MSCs).

## 2. Materials and Methods

### 2.1. Creation of ON Damage in a Rabbit Model

Eight 3-month-old female New Zealand Oryctolagus cuniculus rabbits (Harlan Laboratories Srl, San Pietro al Natisone, Italy) housed at the interdepartmental animal facility of the University of Modena and Reggio Emilia were used. The study protocol was approved by the local animal ethics committee and by the Italian Ministry of Health, according to Replacement, Reduction and Refinement rules [13]. The first part of the study aimed to identify the appropriate approach for the induction of ON in the selected animal model.

The first approach consisted of one intramuscular injection of 20 mg/kg MPSL (Pfizer, New York, NY, USA), which is a known corticosteroid used for ON induction (*n =* 2) [14,15]. 

The second approach was the combination of surgical technique and MPSL (*n =* 2). Each rabbit was sedated with an intramuscular injection of 1 mg/kg midazolam and anesthetized with 3% Sevorane (Abbott S.r.l., Roma, Italy); then, a lateral skin incision was made over the greater trochanter, the subcutaneous tissue was incised, and the tensor fascia latae was split open. The gluteus medius muscle was detached from the greater trochanter and elevated anteriorly, and a “T” cut was made on the anterior part of the hip capsule. Electrocoagulation was performed all around the neck, extending from the greater trochanter to the cartilage on the FH that was left untouched. The ligament of teres was then cut to ensure that no extra-osseous circulation to the FH remained [16,17]. The same procedure was performed on the contralateral hip and 20 mg/kg MPSL was intramuscularly injected. Antibiotic (0.25 g/kg BAYTRIL, Bayer HealthCare LLC Animal Health Division, Pittsburgh, PA, USA) was administered as prophylaxis [18]. 

In both approaches, animals were euthanized 4 weeks after damage induction with a lethal dose of anesthetic, and the FHs were removed and fixed in formalin for subsequent histological assays. These results allowed identifying the combination of surgical technique and MPSL as the most suitable method to generate an ON event with pathogenic features mimicking those reported in humans. Having chosen the approach to induce ONFH, we reproduced the selected method bilaterally in rabbits to observe ON development at 2 weeks (*n =* 2) and at 6 weeks (*n =* 2). After animal sacrifices, FHs were collected for histological assays to identify the proper timing of BM-MSC treatment. The study has been performed as reported in Figure 1.

### 2.2. Autologous BM-MSC Treatment in the ON Rabbit Model

Three-month-old New Zealand female rabbits (*n =* 5) underwent BM harvesting for MSC isolation under general anesthesia. BM blood was collected from the right femoral condyle using a Jamshidi 16G T-type BM aspiration needle (CareFusion, San Diego, CA, USA) connected to a 10 mL syringe pre-loaded with 0.2 mL of Heparin (25,000 IU/mL; Hospira Italia Srl, Napoli, Italy) [19,20,21].

As shown in Figure 1, one week after BM harvesting, animals underwent bilateral ONFH induction as previously described. Two weeks post-ON induction, autologous BM-MSCs were transplanted only in the right FH, maintaining the left FH as a control. Under general anesthesia, a 2 cm lateral skin incision was made over the greater trochanter, and a hole was created using a sharp bone perforator. An intravenous cannula (16G; BD, Becton-Dickinson, Franklin Lakes, NJ, USA) was inserted approximately 2.5 cm through this hole toward the FH for the injection of 2 × 10^6^ BM-MSCs. The hole was closed using bone wax (Johnson & Johnson, New Brunswick, NJ, USA), and the tissue layers were appropriately closed by suture. During MSC transplantation, we harvested BM from the left femoral condyle for MSC isolation to compare the MSC performance before (ON-BM-MSCs) and after the ON damage (ON+BM-MSCs). Two weeks post-implantation, MSC transplanted FHs were removed for histological assays to evaluate the therapeutic efficacy of BM-MSCs in comparison to not transplanted FHs as controls.

### 2.3. BM-MSC Isolation and Amplification

BM samples (*n* = 5) were collected from the femoral condyle and processed as described [22,23]. After treatment with lysis buffer (0.15 M ammonium chloride, 0.009 M potassium bicarbonate, and 0.01 mM EDTA; all from Carlo Erba, Milano, Italy), the samples were centrifuged at 306× *g*; then, cells were counted by 0.4% trypan blue exclusion (Biochrom AG, Berlin, Germany) and inoculated into culture flasks (Greiner Bio-One GmbH, Kremsmünster, Austria). A total of 1 × 10^6^ cells/cm^2^ were seeded and cultured with Minimum Essential Medium Alpha (α-MEM) without nucleosides (Gibco^®^ Invitrogen, Waltham, MA, USA), supplemented with 8% human platelet lysate (PL) [24], 1% L-glutamine (Gibco^®^ Invitrogen), 1 IU/mL heparin (Sigma-Aldrich, St Louis, MO, USA), and 10 µg/mL ciprofloxacin (HIKMA Pharmaceuticals, London, UK).

Nine days post-seeding, fibroblastic colony-forming units (CFU-Fs) were detected, and clones of more than 50 cells were counted on an inverted microscope (Axio Observer A1 with Plan-NeoFluar 10× objective). Cells were kept in incubators with a controlled atmosphere (5% CO_2_, 37 °C) and the medium was replaced every 2–3 days, discarding non-adherent cells.

Once 80–90% confluence was reached, the BM-MSCs were detached using 0.05% trypsin/0.02% EDTA (Gibco), counted, and seeded at 6000 cells/cm^2^. Once BM-MSCs reached passage 3 (P3), cells were detached, and 2 × 10^6^ BM-MSCs were suspended in 100 µL of normal saline (S.A.L.F. SpA, Laboratorio Farmacologico, Bergamo, Italy) and loaded in a 1 mL tuberculin syringe (B. Braun, Melsungen, Germany) for transplantation. Cultures were also maintained until P5 to calculate the cumulative population doubling (CPD) using the following formula (1):PD = log_10_(*Pi* + 1/*Pi*)/(log_10_2)(1)
where *Pi* is the number of cells in each passage in vitro [25].

### 2.4. Immunophenotypic Characterization by Flow Cytometry

When ON-BM-MSCs (*n =* 5) and ON+BM-MSCs (*n =* 5) reached P3, fluorescence-activated cell sorting (FACS) was performed as previously reported [22]. Cells were stained with unconjugated anti-rabbit CD45 and unconjugated mouse anti-human CD29 (all from AbD Serotec, Oxford, UK) in PBS with 0.1% bovine serum albumin (Sigma-Aldrich). Primary antibodies were detected using a goat anti-mouse antibody conjugated with APC (BD Pharmingen) to detect CD29 and conjugated with PE (BD) to detect CD45. The analyses were performed using a FACSAria III flow cytometer equipped with FACS DIVA software (BD).

### 2.5. Differentiation Assays

Once they reached P3, BM-MSCs isolated from both groups (*n =* 5 per group) were induced toward the adipogenic lineage. Cells were seeded at a density of 6000 cells/cm^2^, and at 80–90% cell confluence, cells were incubated with adipogenic induction medium consisting of α-MEM containing 8% PL, 1% P/S, and 1% glutamine supplemented with 10% rabbit serum (Euroclone S.p.A., Milano, Italy), 5% horse serum (Hyclone Laboratories Inc, Logan, UT, USA), 1 µM dexamethasone, 60 µM indomethacin, 10 µM insulin, and 0.5 mM 3-isobutyl-1-methylxanthine (all from Sigma-Aldrich) [26]. BM-MSCs were maintained in differentiation medium for 10 days and stained with Oil Red O solution (Sigma-Aldrich). Adipogenic differentiation was evaluated by the appearance of characteristic clusters of cells containing lipid vacuoles stained in red. Images were acquired on an Axio Observer A1 microscope equipped with Axiovision 4.82 software (Zeiss, Jena, Germany).

Both specimens were also induced toward the osteogenic lineage once they reached P3. Cells were seeded at a density of 6000 cells/cm^2^ into 6-well multiwell plates previously coated with 0.1% gelatin (Gelatin from bovine skin, Type B, Sigma-Aldrich). At 80–90% cell confluence, cells were incubated with bone induction medium consisting of α-MEM containing 8% PL, 1% P/S, and 1% glutamine and supplemented with 10 nM dexamethasone, 10 mM β-glycerophosphate, and 0.1 mM L-ascorbic acid-2-phosphate (all from Sigma-Aldrich). Seven days after the start of the induction, the medium was additionally supplemented with bone morphogenetic protein-2 (100 ng/mL; PeproTech, Rocky Hill, NJ, USA), and on the 14th day, differentiated BM-MSCs and controls were stained with Von Kossa dye [27,28]. Dark deposits of hydroxyapatite were detected using an inverted microscope with 10× magnification (Axio Observer A1 equipped with Axiovision 4.82 software; Zeiss).

### 2.6. Histology

FHs collected from rabbits (*n =* 5 per group) were fixed in 10% neutral-buffered formalin for 2 days, decalcified in fast decal solution composed by 5% formic acid (Sigma-Aldrich) and 5% trisodium acetate (BD) in ddH_2_O for 21 days, and then paraffin embedded [29,30]. Three µm-thick sections were stained with hematoxylin and eosin (H&E, Carlo Erba) [31] and then examined on a Zeiss Axioskop (Zeiss). Micrographs were acquired with an AxioCam ICc3 color camera (Microlmaging GmbH, Carl Zeiss Group, Jena, Germany) and analyzed with AxioVision LE software (Carl Zeiss Imaging Solution, Zeiss). In only FHs collected from animals transplanted with ON-BM-MSCs and their controls, we quantified fat degeneration represented by adipose cysts. Ten micrographs were randomly acquired at 20× magnification for both ON-BM-MSC–treated (*n =* 5) and control specimens (*n =* 5). For each picture, the percentage of adipose areas was quantified by Image J software (National Institutes of Health, Bethesda, MD, USA) using the function Analyze Particles. We also investigated differences in direct or indirect/endochondral osteogenesis by Alcian blue Hematoxylin and Orange G/Eosin staining [32]. Six µm-thick paraffin sections were stained with 1% Alcian blue dissolved in Mayer’s Acid Hematoxylin and counterstained with 2% Orange G and 1% Eosin solution (all Sigma-Aldrich). Micrographs were acquired as previously described.

Based on the properties of endochondral ossification, which is characterized by a higher osteocyte density compared to mature bone [33,34], we introduced a scoring system for osteocyte density (OD) and empty lacuna density (ELD) using the following Formulas (2) and (3):OD = number of osteocytes/bone area(2)
ELD = number of empty lacunae/bone area(3)

Five micrographs were randomly acquired at 20× magnification for both ON-BM-MSC-treated and control specimens. For each picture, bone tissue areas (µm^2^) were quantified using Axiovision 4.8.2 software (Carl Zeiss Imaging Solution), and osteocytes and empty lacunae were counted by two double-blinded operators. OD and ELD were then calculated.

### 2.7. Statistical Analysis

All data collected are presented as mean ± standard error of the mean (SEM). Statistical analysis was performed using paired Student’s *t*-tests. The *p*-values < 0.05 were considered statistically significant.

## 3. Results

### 3.1. Surgical Procedure Combined with MPSL Is More Efficient Than MPSL Alone at Reproducing ONFH in Rabbits

We investigated the ability of MPSL administration, alone or in combination with a surgical procedure, to induce ON damage to identify the best method to obtain an in vivo model that reproduces human ONFH (Figure 2A). At 4 weeks post-induction, cartilage tissue was preserved in both approaches. In the BM compartment, lipidic cysts, fibrin coagula, and inflammatory cells were equally detected in FHs from rabbits subjected to both methods. Very interestingly, in samples from rabbits undergoing the combinatory procedure, we observed a consistent reduction in hematopoietic cellularity. Moving to the bone compartment, the MPSL method did not substantially damage the bone tissue. In contrast, the combination of the surgical procedure and steroids severely affected bone, as shown by the relevant number of pyknotic osteocytes and empty lacunae (Figure 2A).

### 3.2. Two Weeks after ON Induction Is the Best Timing for MSC Implantation to Recapitulate Pathogenic Features

We selected the combination of the surgical procedure and corticosteroids as the most suitable method to generate an ON animal model with pathogenic features observed in humans. In all specimens, we observed the typical histological features of ON [35,36]: fat degeneration, lipidic cysts, reduction in hematopoietic cellularity, fibrin coagula, inflammatory cells, polymorphonucleated cells and macrophages, with empty lacuna.

To evaluate the best timing for MSC implantation, we studied the development of the disease in rabbits at 2 and 6 weeks post-ON induction. As shown in Figure 2B, cartilage was preserved at 2 weeks post-damage. In the BM compartment, we observed adipose hypertrophy, inflammatory cells surrounding adipocytes, and a significant reduction in hematopoietic cells. Numerous empty lacunae and pyknotic osteocytes were detected in the bone tissue, suggesting that ONFH was already developed.

As seen in Figure 2C, cartilage was preserved at 6 weeks post-damage. The BM compartment was characterized by adipocyte hypertrophy, inflammatory cell persistence, and an early restoration of hematopoietic cells. Bone tissue was still damaged, as demonstrated by the empty osteocyte lacunae, but supported by some areas of spontaneous bone regeneration. These areas were characterized by high osteoblast cellular density along the necrotic trabecula; they were slowly reabsorbed and simultaneously replaced with new viable bone. This process seems to be similar to the “creeping substitution” described in humans [37,38]. These results led us to implant ON-BM-MSCs at 2 weeks after ON induction and collect FHs at 2 weeks after MSC implantation. Therefore, the histological evaluation was performed at 4 weeks after ON damage, allowing us to attribute bone regeneration to MSCs despite creeping substitution.

### 3.3. BM-MSCs Improve Osteogenesis into Damaged FHs

We transplanted autologous ON-BM-MSCs into FHs 2 weeks after damage induction to assess the capacity of BM-MSCs to generate new bone.

In the control group, we observed lipidic cysts as well as small elongated and irregularly distributed fibroblast stromal elements mixed with dispersed hematopoietic cells. Moving from the BM compartment to the bone tissue, we detected some osteocytes with pyknotic nuclei and more empty lacunae. Close to the necrotic bone, we observed small areas of viable tissue with a rime of osteoblasts resembling creeping substitution (Figure 3A).

In contrast, in BM-MSC-transplanted FHs, the BM tissue appeared populated with more heterogeneous hematopoietic cells without adipose hypertrophy. Moreover, cuboidal and fibroblast stromal elements were homogeneously distributed. The area of lipidic cysts was quantified by Image J software revealing a percentage of 45.69 ± 6.44% in the control group and 26.81 ± 7.46% in the MSC-treated group. This difference was statically significant (*p* < 0.001; Appendix A, Appendix A). Surprisingly, only the transplanted FHs exhibited a peculiar ossification mimicking endochondral bone regeneration, which was characterized by a matrix densely populated by osteocytes (Figure 3A).

We performed Alcian blue Hematoxylin, Orange/Eosin staining to confirm the endochondral process. As reported in Figure 3B, the control group showed only a homogeneous orange/pink staining identifying the presence of bone tissue, and pyknotic osteocytes and empty lacunae were still observed. Interestingly, only in the BM-MSC-transplanted group, close to the necrotic area, were blue positive shades detected within orange/pink bone matrix and a blue ring surrounded the osteocytes, emphasizing their metabolic activity (Figure 3B).

In accordance with the endochondral process, large areas of high osteocyte density were observed in treated FHs compared to controls. In order to evaluate any changes in the bone compartment, empty lacunae and new osteocytes were quantified. As seen in Figure 3C, the BM-MSC treatment statistically significantly increased the OD (1831.32 ± 189.62) compared to the ELD (516.92 ± 61.39; *p* < 0.05). No significant differences in OD (1138.64 ± 111.19) and ELD (810.79 ± 87.84; *p* > 0.05) were detected in the controls. In addition, we observed a statistically significant increase in OD in MSC-treated FHs compared to controls (*p* < 0.05). In conclusion, BM-MSCs can improve osteogenesis into damaged FHs early after transplantation.

### 3.4. MSCs Isolated from FHs after Damage Show a Decrease in Clonogenic Ability

Having defined the appropriate ONFH rabbit model, we investigated the impact of ON damage on the BM microenvironment by comparing ON-BM-MSCs and ON+BM-MSCs.

BM samples were collected from the right femoral condyle before ON induction (2.25 ± 0.17 mL) to isolate ON-BM-MSCs and from the left femoral condyle post-ON damage (3.45 ± 0.26 mL) to isolate ON+BM-MSCs. The difference in BM volume suggested an increase in liquid in the BM compartment after the ON event, although no statistically significant difference was observed (*p* > 0.05, data not shown).

From the BM specimens, we isolated and counted 4.88 ± 3.03 × 10^8^ and 6.23 ± 1.26 × 10^8^ mononucleated cells, respectively, for the ON-BM-MSC and ON+BM-MSC specimens (Figure 4A). Four days after seeding, fibroblastoid elements began to adhere to the plastic surface, and after 9 days of culture, CFU-Fs were detected (Figure 4B) [39].

The mean CFU-Fs was significantly higher in the ON-BM-MSCs (29.03 ± 3.61/10^7^) compared to the ON+BM-MSCs (6.94 ± 0.97/10^7^; *p* < 0.01; Figure 4B). In Figure 4C, we reported the similar cell morphology of both specimens during culture from P1 to P5. We observed an increase in cell size with evident cytoskeleton at P5 in both samples.

The CPD was similar for ON-BM-MSCs and ON+BM-MSCs, showing a gradual increase until P4 (6.88 ± 0.62 for ON-BM-MSCs vs. 5.43 ± 0.72 for ON+BM-MSCs). Interestingly, at P5, the CPD of ON-BM-MSCs decreased (5.39 ± 1.87), while the CPD increased for ON+BM-MSCs (6.68 ± 0.23), as shown in Figure 4C (*p* > 0.05).

### 3.5. ON-BM-MSCs and ON+BM-MSCs Exhibit the Same Immunophenotypic Profile and Differentiation Potential

We then examined cell populations using immunophenotypic characterization (Figure 5A) and differentiation assays (Figure 5B) to assess the presence of other cell types, such as monocytes and endothelial cells, that could adhere to the plastic surface.

Flow cytometry analyses were performed as previously described by Piccinno et al. and limited to CD29 and CD45 antibodies [40]. As seen in Figure 5A, ON-BM-MSCs and ON+BM-MSCs were weakly positive for CD45 (8.84 ± 2.83 and 13.25 ± 2.80, respectively), with the observed difference suggesting an increase in immune cell number in ON+BM-MSCs (*p* > 0.05). CD29 was equally expressed in ON-BM-MSCs and ON+BM-MSCs (65.44 ± 4.89 and 67.31 ± 6.51, respectively).

After adipogenic induction, we observed round lipid droplets inside the cells, and Oil Red O staining confirmed the adipose differentiation in both MSC populations (Figure 5B). Similarly, after osteogenic induction, Von Kossa staining was performed, and calcium deposits were detected at high levels in induced samples, but they were absent in un-induced controls (Figure 5C). In conclusion, the proliferation rates, immunophenotypic profiles, and differentiation abilities were comparable in the ON-BM-MSCs and ON+BM-MSCs, suggesting that ON did not impact the BM microenvironment.

## 4. Discussion

The current treatments for ONFH are not always effective in preventing the bone architecture collapse, lastly requiring a total hip replacement [3,41]. The identification of advanced clinical treatments represents an urgent medical need.

In this study, we implanted autologous BM-MSCs into an induced ON rabbit challenge model. For the first time in our knowledge, we proposed an alternative animal model able to recreate the pathology through the same steps described for humans.

ON could be caused by both non-traumatic and traumatic events. Traumatic ON has a low incidence in humans compared to non-traumatic ON, which is commonly caused by the administration of corticosteroids [42]. The most common strategy described in the literature to generate animal models of ONFH is the injection of MPSL, combining different doses and number of administrations [43,44]. In our study, we proposed a synergic role of corticosteroid administration and surgical procedure in order to obtain a complete histological pattern that includes all pathogenic features: empty lacunae, lipidic cysts, reduction in hematopoietic cellularity in the BM, fibrin coagula, inflammatory cells, polymorphonucleated cells and macrophages.

For the first time, to our knowledge, we identified the combination of surgery and MPSL as the optimal approach to generate ONFH in rabbits, better inducing fat degeneration, depletion of hematopoietic cells and bone tissue necrosis compared to corticosteroid alone.

The identification of a proper animal model of ONFH then allows the assessment of new therapeutical approaches, such as an MSC-based strategy. BM-MSCs are generally administered intravenously or locally at the damaged site [43,44]. The debate on the intravenous administration of MSCs incorporates several opposing theories. The possibility of an accumulation of MSCs in the lung is not to be excluded, raising questions about its efficacy in therapeutic approaches where cells have to reach a specific location [45].

In this study, we administered MSCs locally rather than systemically, in a sufficient number aiming to achieve a therapeutic outcome, as previously suggested by others [46]. After BM-MSC implantation, we proceeded with the removal of the FHs and the subsequent histological assays. In vivo experiments lasted only 2 weeks to limit spontaneous bone regeneration and thus enable evaluation of the real contribution of implanted BM-MSCs in osteogenesis. MSC-transplanted animals showed an improvement in features compared to control animals, such as a repopulated BM compartment, a significant reduction in adipose hypertrophy and regenerated bone tissue through endochondral ossification.

Indirect ossification is a type of physiological bone formation that occurs during embryonic, fetal, and early postnatal life from a template of cartilage [33,47]. It is a prerequisite for the formation of the niche with a subpopulation of hematopoietic progenitors that form fetal bone with a medullary cavity [48]. This might also explain why there was a greater hematopoietic cellularity in treated samples compared to controls.

Alcian blue hematoxylin and eosin/orange G staining confirmed the endochondral process close to necrotic trabecula in rabbits transplanted with autologous MSCs, while in the control group, the few areas of bone regeneration were Alcian blue negative, suggesting a direct ossification [49]. This observation provides further evidence that MSCs play a pivotal role in bone regeneration in the ONFH model.

Finally, we introduced a scoring system based on ELD and OD parameters as further confirmation of the efficacy of the treatment. OD is a known parameter used to evaluate osteogenesis that correlates with the biomechanical quality of the bone in humans as reported in the literature [34,50]. In this study, the 60% increased OD in the treated group supports the potential of autologous implanted BM-MSCs in skeletal regeneration for this ON model.

One of the limitations of current preclinical models is related to the use of BM-MSCs collected before the ON induction, from the healthy microenvironment, which omits the effect of disease on MSC performance. This is an issue for translation toward the clinic, where autologous MSCs are collected from patients with ON. Therefore, to overcome these shortcomings, we isolated BM-MSCs before (ON-BM-MSCs) and after ON induction (ON+BM-MSCs) and compared them in terms of clonogenic ability, growth rate, immunophenotypic profile, and ability to differentiate toward the adipose and bone lineages.

This approach additionally enabled us to study the impact of ONFH on BM-MSC performance in an autologous setting by comparing rabbit ON-BM-MSCs and ON+BM-MSCs.

Our study revealed a significant increase in BM volume harvest in ON+ compared to ON-FHs. While this has not been completely clear, we can speculate that the infarction of bone tissue and the subsequent inflammatory reaction leads to liquid accumulation. This assumption is supported by the observation of a greater number of mononucleated cells/mL of BM for ON+ than ON- rabbits but a significant (30%) reduction in CFU-F number in ON+ compared to ON- samples. This could also suggest the higher presence of immune cells in ON+ BM diluting the number of MSCs able to form CFU-Fs. Thus, we can assert that ON damage impacts the BM microenvironment, although a cell fraction able to generate CFU-Fs is preserved. Comparing MSC in vitro performance, we observed a similar growth rate of ON-BM-MSCs and ON+BM-MSCs. We hypothesize that the disease exerted a selective pressure on the MSC population, leading to a decline in CFU-F number but allowing the survival of the best-performing clones.

Flow cytometry analysis was performed to discriminate CD29 positive rabbit MSCs from CD45 positive white blood cells. ON-BM-MSCs and ON+BM-MSCs showed a similar expression of CD29 (>65%) while CD45 was about 50% more expressed in ON+BM-MSC compared to ON-BM-MSC samples, which is a further confirmation of white blood cells as observed in CFU-F assay [51,52,53].

Furthermore, focusing on ON-BM-MSCs, we noticed a better growth rate until P5 than those reported in the literature for rabbit MSCs [54]. We can attribute this difference to the culture medium used for the MSC maintenance. Although Dulbecco’s modified eagle medium (DMEM) is described in the literature as the typical MSC basal culture medium, we selected α-MEM to increase MSC proliferation and clonogenicity, as reported by Lapi et al. [55]. In addition, we replaced 10% fetal bovine serum (FBS) with 8% human PL in rabbit MSC culture [55] in order to overcome the limits of FBS variability among lots.

The comparison between ON-BM-MSCs and ON+BM-MSCs demonstrated that the ON microenvironment did not affect MSC performances. This aspect seems particularly relevant for the clinical translation of autologous BM-MSC implantation.

To our knowledge, other alternative therapeutic treatments are also being developed and are currently in the pre-clinical phase, as reported in the literature [41,56]. Asada et al. treated a rabbit model of induced ONFH with autologous BM and observed an improvement in treated animals compared to control ones. They attributed the beneficial effect to the fraction of MSCs within the BM [57]. In fact, other investigators used MSCs combined with ceramic scaffolds in a dog model [58], showing their potential efficacy in ONFH treatment. Gene therapy for ONFH is another approach based on the implantation of modified MSCs into the necrotic area of ONFH to stimulate angiogenesis and cell proliferation for bone regeneration [9,59,60,61]. The strength of our approach is based on the use of a not genetically modified, autologous, and selected MSC population.

In this study, we originally introduced a rabbit model faithfully mimicking the development of human ON features to then introduce a therapeutical approach based on autologous MSC as a regenerative medicine tool potentially able to reduce, postpone or abrogate prosthetic strategies in humans.

## Figures and Tables

**Figure 1 pharmaceutics-14-02127-f001:**
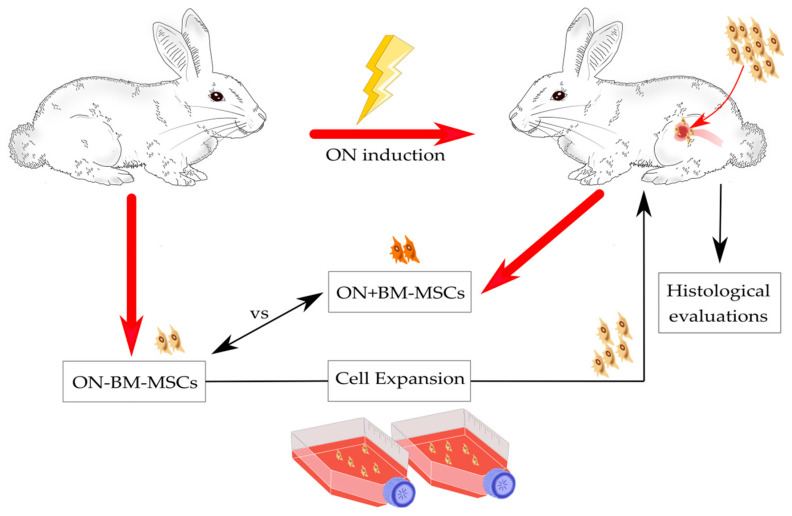
Overview of the experiments. Bone marrow (BM) was harvested from five rabbits and mesenchymal stromal/stem cells (MSCs) were isolated and expanded in vitro. Bilateral osteonecrosis (ON) damage was created in femoral heads (FHs), and after two weeks, BM was again harvested in order to isolate MSCs and compare cells before (ON-BM-MSCs) and after (ON+BM-MSCs) ON induction. In parallel, ON-BM-MSCs were implanted into the damaged FHs. Two weeks post-implantation, FHs were harvested and analyzed by histological assays to evaluate bone regeneration.

**Figure 2 pharmaceutics-14-02127-f002:**
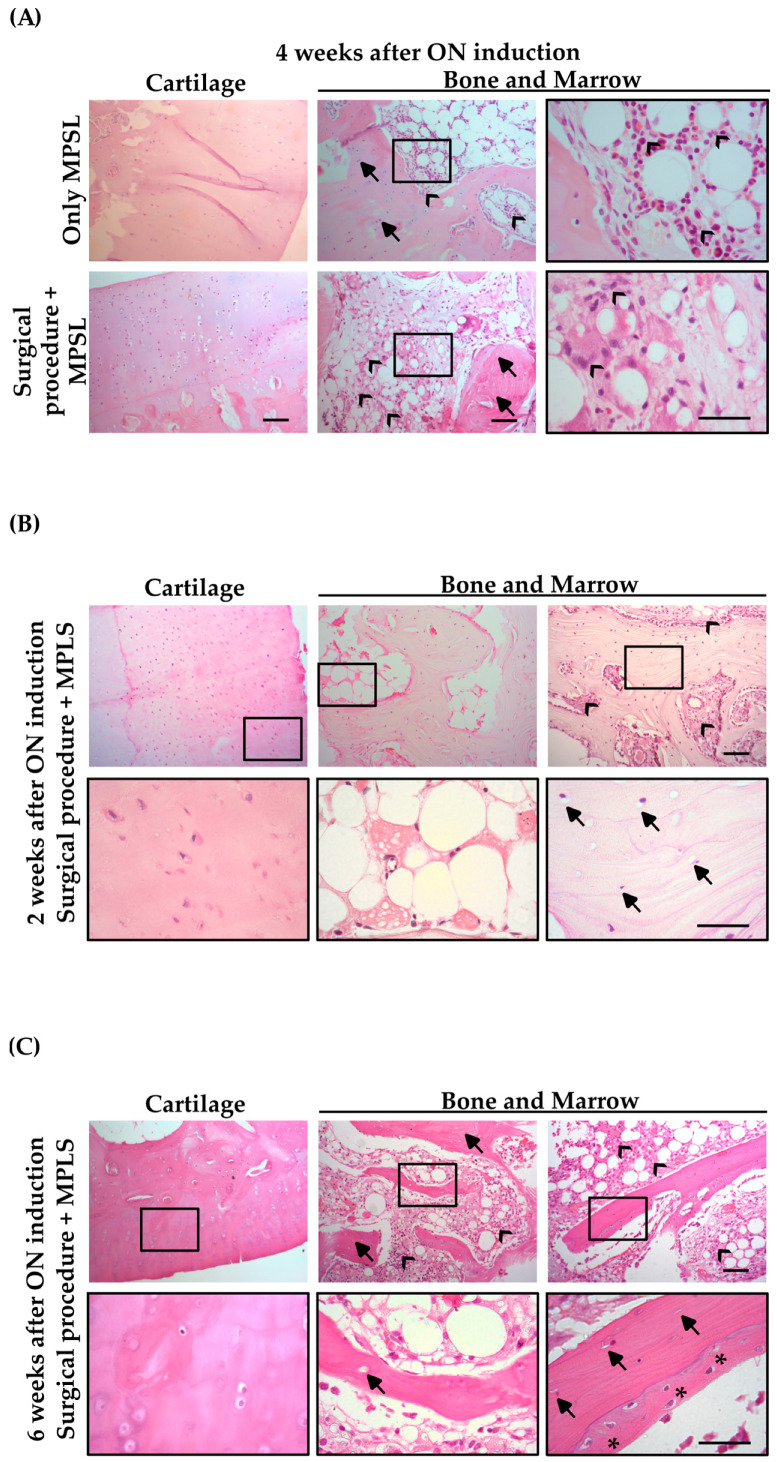
Histological analyses at different times after the osteonecrosis (ON) damage induction. (**A**) Comparison at 4 weeks after the ON event using only the pharmacological method with methylprednisolone (MPSL; upper panel) or the combination of the surgical procedure and MPSL (bottom panel), depicting cartilage and bone tissue. (**B**) The combination of surgical procedure and corticosteroid was also analyzed at 2 weeks and (**C**) 6 weeks after ON damage (10× and 40× magnification; scale bar 100 µm and 50 µm. Arrows: osteocyte lacunae; asterisks: spontaneous bone regeneration; arrow heads: hematopoietic cellularity).

**Figure 3 pharmaceutics-14-02127-f003:**
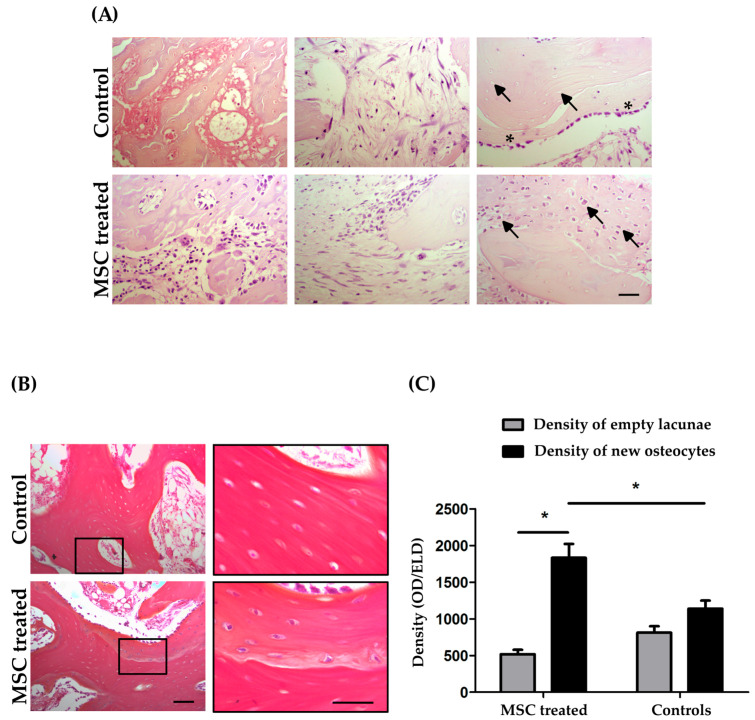
Histological analyses of femoral head (FH) treated with mesenchymal stromal/stem cells (MSCs) compared to the control. (**A**) In the MSC-treated group, there is a more populated bone marrow and a better organization of fibroblast stromal elements compared to control. A reactive osteoblastic rim is present in the control rabbit specimens as well as in the transplanted ones, but only the treated femoral head (FH) shows a similar endochondral ossification (10× magnification; scale bar 100 µm). (**B**) The Alcian blue staining for bone maturity reveals a new bone tissue in treated FHs (10× and 40× magnifications; scale bar 100 µm and 50 µm). (**C**) Osteocyte density evaluation in terms of empty lacunae (ELD) and new osteocytes (OD) in control and treated rabbits. (* *p* < 0.05; arrows: osteocyte lacunae; asterisks: spontaneous bone regeneration).

**Figure 4 pharmaceutics-14-02127-f004:**
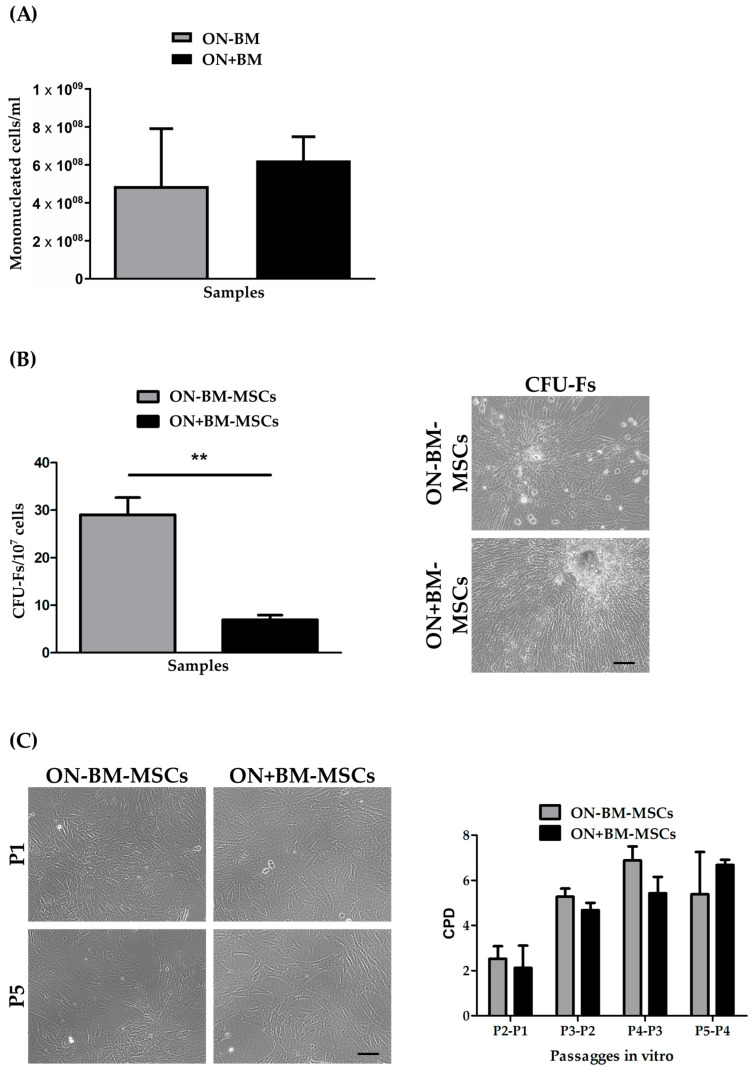
Mesenchymal stromal/stem cell (MSC) isolation from bone marrow (BM). (**A**) Mononucleated cells per ml of BM from rabbit femoral head before and after osteonecrosis (ON) damage. (**B**) The fibroblastic colony-forming units (CFU-Fs) per 10^7^ cells were counted 9 days after seeding for MSCs isolated from BM before (ON-BM-MSCs) and after (ON+BM-MSCs) ON event (** *p* < 0.01; 10× magnification; scale bar 100 µm). (**C**) Cell morphology of ON-BM-MSCs and ON+BM-MSCs (left panel, 10× magnification; scale bar 100 µm) and their cumulative population doubling (CPD) until passage 5.

**Figure 5 pharmaceutics-14-02127-f005:**
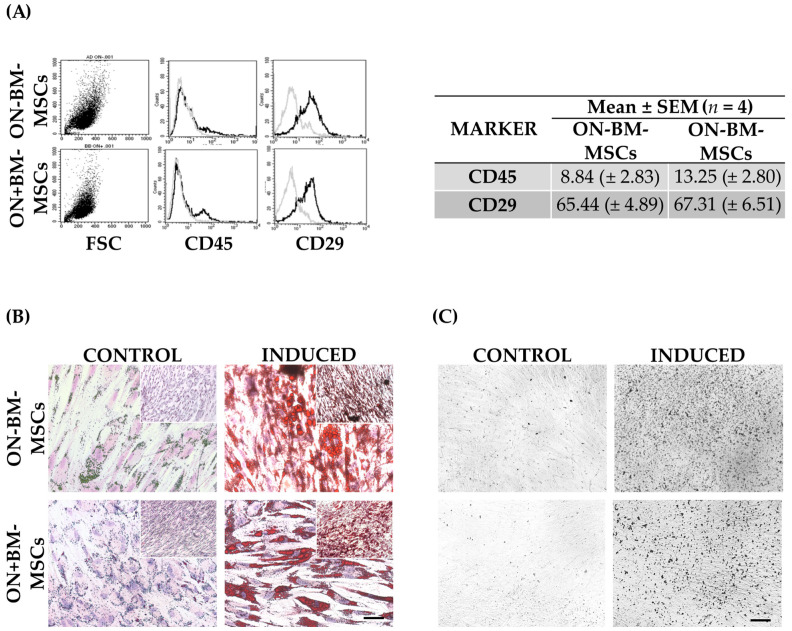
In vitro characterization and comparison of bone marrow-derived mesenchymal stromal/stem cells isolated before (ON-BM-MSCs) and after (ON+BM-MSCs) ON event. (**A**) Immunophenotypic profile using CD45 and CD29 antibodies and their expression in the two cell populations as mean ± SEM. (**B**) Oil red O staining to detect adipogenic in vitro differentiation of two MSC populations (10× in inset and 20× magnifications; scale bar 100 µm and 50 µm) and (**C**) Von Kossa staining to detect osteogenic in vitro differentiation of two MSC populations (10× magnification; scale bar 100 µm).

## Data Availability

Not applicable.

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
