# Peer review of "Autologous Marrow Mesenchymal Stem Cell Driving Bone Regeneration in a Rabbit Model of Femoral Head Osteonecrosis"

_pharmaceutics, 2022, doi:10.3390/pharmaceutics14102127_

Round 1

Reviewer 1 Report

·         How many BM-MSCs were implanted?

·         Did the authors expand BM-MSC before implant? Or they just froze in liquid nitrogen?

·         Line 209-216, please indicate hematopoietic cells and cartilage tissue in Figure 2.

·         Line 232, please indicate hematopoietic cells.

·         The author used anti-human CD29 to indicate MSC. However, they applied goat anti-mouse antibody to detect CD29. Please clarify.

·         Please add more surface markers such as CD73, CD90, and CD105.

·         Please reorganize Figure 2A as 2B and 2C.

·         Line 252, what’s the “control group” indicate? Health or surgical ON? When discussing ON-BM-MSCs, are they indicated as the same group?

·         What’s the mechanism of implanted MSCs? How does it facilitate osteogenesis?

·         Line 288, please change the font of “4.88 ± 3.03 x 108 and 6.23 ± 1.26 x 108 mononucleated cells”

·         Line 388-389, “Finally, we introduced a scoring system based on ELD and OD parameters as further confirmation of the efficacy of the treatment.” what’s the score of the current experiment?

Reviewer 2 Report

Although the underlying premise of the study was interesting, were multiple concerns with its execution and presentation:

1. the methods section was particularly confusing. It was difficult to discern how many animals ended up in the comparison at each step. It appears like for the first step (comparing methods of ON induction), there were two animals per group, a number too small to draw statistically significant difference. Indeed the comparisons in fig2 seemed to lack quantification and were all subjective. It was unclear what the authors meant by "new osteocytes", given that new bone formation dynamics were not quantified. The term "density" is also very confusing, given that bone mineral density was not studied. It was not clear why bone histology was studied at 2 and 6 weeks and the comparison data were not outlined, or perhaps missed by the reviewer. 

2. There was little quantification throughout the paper. It was unclear how many repeats were done for in-vitro studies. Figure 4C had no error bars. 

3. Bone adiposity and vascular perfusion were not quantified

Round 2

Reviewer 1 Report

The Authors have addressed all of my concerns.